# Muscle Contractile Properties Measured at Submaximal Electrical Amplitudes and Not at Supramaximal Amplitudes Are Associated with Repeated Sprint Performance and Fatigue Markers

**DOI:** 10.3390/ijerph182111689

**Published:** 2021-11-07

**Authors:** Alejandro Muñoz-López, Moisés de Hoyo, Borja Sañudo

**Affiliations:** 1Department de Motricidad Humana y Rendimiento Deportivo, University of Seville, 41927 Seville, Spain; 2Department of Physical Education and Sports, University of Seville, 41927 Seville, Spain; dehoyolora@us.es (M.d.H.); bsancor@us.es (B.S.)

**Keywords:** tensiomyography, fatigue, creatin kinase, peak torque, repeated sprint performance

## Abstract

Background: The present study analyzes the associations between the muscle contractile properties (MCP) measured at different neuromuscular electrical stimulation amplitudes (NMESa) and the performance or transient fatigue after a bout of repeated sprints. Methods: Seventeen physically active male subjects performed six repeated sprints of 30 m with 30 s of passive recovery. Capillary blood creatine kinase (CK) concentration, knee extension or flexion isometric peak torque, tensiomyography, and repeated sprint performance were assessed. Results: Muscle displacement and contraction time were different in relation to the NMESa used in the rectus femoris and biceps femoris muscles. At rest, significant (*p* < 0.05) associations were found between muscle displacement and the loss of time in the repeated sprints (sprint performance) at 20 or 40 mA in the rectus femoris. At post +24 h or +48 h, the highest significant associations were found between the muscle displacement or the contraction time and CK or peak torques also at submaximal amplitudes (20 mA). The NMESa which elicits the peak muscle displacement showed lack of practical significance. Conclusion: Although MCP are typically assessed in tensiomyography using the NMESa that elicit peak muscle displacement, a submaximal NMESa may have a higher potential practical application to assess neuromuscular fatigue in response to repeated sprints.

## 1. Introduction

Tensiomyography (TMG) is a valid and reliable [1] method to measure muscle contractile properties (MCP). TMG can instantly determine the MCP in response to different muscle actions [2]. It uses an external electro stimulator to evoke muscle fiber activation, thus estimating the level of the motor units activated [3] using fixed neuromuscular electrical stimulation amplitudes (NMESa). Typically, the NMESa is increased from lower (i.e., 20 mA) to higher amplitudes (i.e., 100 mA) [4,5]. A commonly used option is to stop the increments when the muscle belly reaches the peak muscle displacement (Dm) [6], which is a consequence of recruiting the maximum possible muscle fibers after the electrical stimulation [7], also known as supramaximal amplitude (DmMax). Dahmane et al. [8] showed that peak Dm assessed at DmMax is strongly related to muscle force. However, Feiereisen et al. [9] showed that a submaximal NMESa (i.e., 10% of DmMax) reflects the recruitment of surface fibers, which is the area where a higher density of fast-twitch fibers is found [10].

Dahmane et al. [8] found that muscles with a lower composition of fast-twitch fibers have a lower contraction time (Tc). Consequently, at DmMax, Travnik et al. [11] confirmed that muscles with more fast-twitch fibers showed a lower Tc and a lower Dm. Furthermore, the authors demonstrated that bed rest negatively affected fast-twitch fibers and, consequently, Tc increased, but Dm decreased [12]. Henneman’s size principle reads that smaller motor units (i.e., slow-twitch fibers) are recruited first [13] during voluntary muscle activations. Nevertheless, Knaflitz et al. [14] suggested that the recruitment order for surface electrical stimulation differs from a voluntary contraction. For instance, Dahmane et al. [7] confirmed that when using an electrical stimulator, such as in the case of TMG, there exists a reverse recruitment order and, in consequence, fast-twitch fibers are activated first. It has been shown that muscles with 40 to 60% of slow-twitch fibers demonstrate an increment in Tc as NMESa increases [15]. Despite the importance of the NMESa on the recruitment pattern, it is not clearly understood how submaximal amplitudes may influence the MCP when using TMG. 

Some authors used different NMESa to assess MCP [7,16]. Morales-Artacho et al. [17] found that Dm increased from 40 mA to 100 mA in the biceps femoris (BF). In contrast, the Vastus Lateralis only showed Dm differences at 40 mA. Dahmane et al. [7] used NMESa of 10% and 50% of DmMax on the biceps brachii. The authors found a non-random fiber-type distribution in the cross-section of skeletal muscles, showing higher correlations between the 50% of DmMax and percentage of slow-twitch muscles, compared to 10% of DmMax. Therefore, the lower the NMESa, the higher the relationship with fast-twitch fibers. These findings could explain the results from Loturco et al. [16], who showed that jumping or sprinting performances were better explained at 40 mA than at other amplitudes. The authors speculated that the best sprinters and jumpers could effectively recruit more motor units under lower amplitudes, even under the size principle compromise [13]. However, and despite the aforementioned evidence, to our knowledge, no other research has studied the potential use of analyzing the MCP at different amplitudes compared to DmMax. 

In sports, there are frequent bouts of fatiguing actions, such as after performing a bout of sprints without recovery (i.e., repeated sprints, RS) [18]. RS increases the muscle peripheral fatigue [19]. Muscle fatigue can be defined as any decline in muscle performance associated with muscle activity [20]. Transient fatigue is notable after fatiguing sports tasks, affecting the electrochemical downstream of the muscle fibers and, in consequence, reducing the contractile muscle capacity [21]. Recently, Muñoz-López et al. [2] showed reductions in the isometric and isokinetic peak torque after fatiguing concentric leg extensions, together with decreases in Dm measured at DmMax. In agreement with these findings, a recent review [22] showed that Dm decreases with fatigue, while Tc increased or remains stable. In addition, Sánchez-Sánchez showed that Tc, but not Dm, changed after a series of RS in soccer players [23], despite that they used the highest stimulation amplitude to analyze data. In a recent meta-analysis, Lohr et al. [1] concluded that robust evidence for diagnostic accuracy and criterion validity of TMG to assess exercise-induced muscle fatigue has yet to be established. Nevertheless, the authors also supported that several validity investigations revealed significant post-intervention relationships between Dm and the reference standard of fatigue (i.e., the maximal voluntary contraction). Consequently, we suggest that submaximal NMESa could provide new insights and validations for the use of TMG to assess MCP.

Given the influence of the NMESa on the selective recruitment of fiber types when using TMG, it is important to study how physical performance and subsequent fatigue may affect the MCP measured at different NMESa. It can be speculated, in line with prior findings [16], that MCP could change according to the NMESa used. Therefore, our objectives were to study (1) the relationship between MCP measured at different NMESa and the performance of RS and (2) the relationship between MCP measured at different NMESa and the transient fatigue exhibited after exercise. Our main hypothesis was that MCP measured at a submaximal NMESa may be better associated with RS performance, and that the associations with transient fatigue would be higher with a submaximal NMESa.

## 2. Materials and Methods

### 2.1. Participants

Seventeen physically active male participants volunteered for this study (mean ± SD: age 23.2 ± 9.8 years, height 1.75 ± 0.01 m, body mass 72.7 ± 21.0 kg). The inclusion criteria were to be physically active and involved on a weekly lower limb resistance training. Individuals were excluded if they had a known cardiovascular, metabolic, or respiratory disease, were unable to perform vigorous exercise, or had a recent lower limb injury in the past 6 months. They were required to refrain from exercise, caffeine, and alcohol for 24 h before testing and to abstain from vigorous exercise throughout the experimental period. Each participant was fully informed about the procedures, potential risks, and benefits of the study, and they all signed a written informed consent form before the tests. The study was conducted following the Declaration of Helsinki and was approved according to the University of Seville’s ethical standards.

### 2.2. Procedures

The participants visited the laboratory on one day for the familiarization (consisting of testing and main intervention exercises) and three subsequent days, interspersed by a week. On the first day, pre- and post-tests were carried out, together with the main intervention. The intervention consisted of six bouts of RS (6 ×·30 m—main intervention [23]), interspersed by 30 s of passive recovery. On the second and the third days, interspersed by 24 h, the participants performed the testing battery. All measurements took part between 9.am and 12.pm in the morning.

The testing battery consisted of measuring the CK activity, the MCP using TMG, and the knee-extension and flexion isometric peak torque. Before any measurements, the participants remained seated for ten minutes. Next, we collected a 30 µL fingertip capillary blood sample for CK determination using a spring-loaded disposable lancet (Safe-T-Pro Plus, Accu-Chek, Roche Diagnostics GmBH, Berlin, Germany). The whole blood sample was immediately pipetted to a test strip and analyzed for CK concentration using a colorimetric assay procedure (Reflotron Plus; Roche Diagnostics, West Sussex, UK). Subsequently, we conducted a TMG test on each limb in the rectus femoris (RF) and BF, following a protocol published elsewhere [2]. Briefly, NMESa started at 20 mA and was increased in ranges of 20 mA every 15 s to avoid electrical fatigue of the motor unit [24], until 100 mA. Tc and Dm (main dependent variables) values for each NMESa on each muscle were recorded for further analyses.

After the baseline measures, the participants performed a standardized warm-up protocol consisting of 3 min of joint mobilization exercises and 5 min of cycling at an easy pace (80 w at 100 rpm) on an electronically braked cycle ergometer (Ergoselect 200, Ergoline, Berlin, Germany). Finally, the participants performed a maximal voluntary isometric contraction test (MVIC) for the knee flexor and extensor muscles in both legs using an isokinetic dynamometer (Biodex System 4, Biodex Medical Systems, Shirley, NY, USA). Each participant remained in a seated position with the seat back tilt angle set to 80 degrees and stabilized with straps at the shoulders, waist, and thighs as per the manufacturer’s guidelines. Participants performed three maximal attempts for each of the isokinetic contractions (knee extension contractions at an angle of 90° and another three knee flexion contractions at an angle of 30°) separated by 60 s. Each repetition lasted 5 s. The peak torque (N·m) was recorded for each muscle action.

After the MVIC determination, participants performed a specific sprinting warm-up consisting of four 20 m running accelerations with progressive intensity with 1 min rest periods between them. Immediately after, they performed the main intervention. Participants remained standing with the lead-off foot placed 1 m away from the first timing gate (with a vertical height of 0.80 m). Then, participants completed the main intervention using four sets of timing gates (Polifemo Radio Light; Microgate, Bolzano, Italy). Standardized verbal encouragements were given throughout the protocol. Sprint time from 0 to 30 m was considered for further analyses. The best RS time trail (RS-Best), the average change between RS trials (RS-Change), the loss in RS time (RS-Loss, as the difference between the first and last time trial) [25], and the RS-Fatigue (REF Borja) were then calculated. After the last sprint, the participants conducted all the testing battery again and repeated it at 24 h [+24 h] and 48 h [+48 h] (Figure 1).

### 2.3. Statistical Analysis

Data from both legs were included in the analyses. Thus, the total sample size was N = 34. Data are shown as mean ± standard deviation. The normality assumption was tested before any statistical test using the Shapiro−Wilk test. First, we tested the between-NMESa differences using a One-Way Repeated Measures ANOVA (time (i.e., Pre or Post) as a simple effect). In the case of sphericity assumption violation, we corrected the data with a Greenhouse-Geisser correction. Furthermore, we performed post-hoc tests with the Bonferroni adjustment for pairwise comparisons. Second, we performed a correlation analysis at Pre between MCP and RS performance or fatigue markers. We also tested the possible relationships (Pearson or Spearman ranking) between changes (Δ) from Pre to consecutive time points (to Post, to +24 h, and to +48 h) in the MCP measured at different NMESa and RS performance or fatigue markers. Only significant correlations were used for further analyses and data interpretation. Third, based on previous analyses, we chose the NMESa that showed the highest prior correlations to analyze time-course changes using a repeated measures design. Thus, 20, 40, 100, and DmMax mA were analyzed. Hence, we tested time (four time points: Pre, Post, +24 h, or +48 h) changes (main factor) using the same procedures as for between-NMESa differences. Finally, in both repeated-measures designs, we analyzed the magnitude of the main effect (ES = effect size), calculating the partial eta-squared (η^2^_p_), in addition to Cohen’s d-effect (d) size statistics for parametric paired t-test comparisons. In addition, we qualitatively analyzed the correlations using the following scale: trivial (<0.1), small (0.1 to 0.3), moderate (0.3 to 0.5), large (0.5 to 0.7), very large (0.7 to 0.9), and extremely large (0.9 to 1.0). For all the analyses, we set the significance level at *p* < 0.05. We used RStudio software (v. 1.2.5033, RStudio: Integrated Development for R. Rstudio, Inc., Boston, MA) with the corrplot package (v. 0.84) to perform the correlation analyses and correlation graphs, and JASP software (v. 0.12.2, JASP Team) for the rest of the analyses.

## 3. Results

### 3.1. Repeated Sprint

RS-Change was 0.27 ± 0.16 s, RS-Best was 4.42 ± 0.20, and there was a mean RS-loss of 0.07 ± 0.05 s over the sprints.

### 3.2. MVIC and Fatigue Time Course

Fatigue markers showed significant changes in time (leg extension MVIC: *p* = 0.004, η^2^*_p_*= 0.127; leg flexion MVC: *p* = 0.003, η^2^_p_ = 0.170; CK: *p* < 0.001, η^2^_p_ = 0.408). More concretely, leg extension MVIC significantly increased from Post to +24 h (*p* = 0.027, d = 0.506) and decreased from +24 h to +48 h (*p* = 0.008, d = 0.572), while leg flexion MVIC was significantly different between Pre and +48 h (*p* < 0.001, d = 0.759). CK significantly increased from Post to +24 h (*p* < 0.001, d = 1.017).

### 3.3. Amplitude Differences 

Between-amplitude differences in BF and RF for each time course measurement are shown in Table 1. There was a significant amplitude effect at all time points for Tc and Dm in both muscles, except for BF-Tc at +48 (*p* = 0.119, η^2^_p_ = 0.056). In the BF, the highest Tc was found at DmMax, while in the RF it was found at submaximal NMESa. These values were consistent before and after the RS protocol, up to +48 h. However, the Dm pattern response, also related to the NMESa, changed after the RS protocol.

### 3.4. MCP Relationships with Fatigue and Performance Markers

Figure 2 shows the relationships between basal MCP at different NMESa and performance or fatigue markers. In BF, there was a large positive association between Tc and CK at high NMESa (100 mA or DmMax). Furthermore, a large negative association was observed between Dm and CK at the lowest NMESa (20 mA). In the RF, there was a moderate negative association between Tc and RS-Loss at low NMESa (20 and 40 mA). In addition, moderate negative associations were found at the lowest NMESa (20 mA) between Dm and several RS performance markers (RS-Change, RS-Loss, and RS-Fatigue). RS-Loss and RS-change showed moderate negative associations at higher NMESa, with the highest association presented at 40 mA.

ΔMCP and Δfatigue marker associations are shown in Figure 3 and Figure 4 for BF and RF, respectively. In BF, a large positive association was found between ΔTc and ΔMVIC at Post, +24 h, and +48 h when the lowest NMESa (20 mA) was used. In this muscle, there were also positive associations between ΔDm and ΔMVIC at Post (large), +24 h (large), and +48 h (moderate). In addition, in BF, at +48 h, there was a large positive association between ΔTc and ΔCK and between ΔDm and ΔCK. The latter also showed a moderate positive association at 100 mA- and DmMax. In RF, positive associations were found at the lowest NMESa (20 mA) between ΔTc and ΔMVIC (very large at Post, large at +24 h and +48 h). Finally, large associations between ΔTc or ΔDm and ΔCK were found at all time points.

Figure 3 and Figure 4 also show the associations of MCP and RS performance markers. In BF, the highest associations were found between ΔTc and RS-change (large), RS-Loss (large), or RS-Fatigue (large) with the highest NMESa (100 mA), but only at +48 h. In RF, the highest associations were found between ΔTc and RS-change (moderate) or RS-Loss (moderate) at high amplitudes but at Post and at +48 h. In this muscle, there were also associations (negative) between ΔDm and RS performance markers. More concretely, at Post, there were moderate associations at 60 mA with RS-change, RS-Loss, or RS-Fatigue, while at +48 h, those associations with RS-change and RS-Loss increased (large) at 40 mA. Finally, the ΔDm and RS-Best were moderately positively associated at higher NMESa at +24 h (100 and DmMaxmA) and at +48 h (80 and 100 mA).

## 4. Discussion

When TMG is used, MCPs are typically analyzed at a supramaximal amplitude (i.e., DmMax). However, confirming our main hypothesis, we observed that the MCPs were related to the NMESa used. Prior to the RS protocol, the MCP showed higher associations with RS performance markers at submaximal NMESa compared to DmMax. Further, these associations were muscle-dependent (more associations found in RF). In contrast to basal levels, submaximal NMESa had a higher relevance (i.e., higher associations) in the presence of fatigue. From all the RS performance markers used, RS-Loss showed more associations with MCP.

From our results, it must be highlighted that when MCPs are analyzed, caution should be taken considering the NMESa used. The majority of studies published using TMG increase the NMESa until the point at which the muscle belly experiences the highest transversal displacement [6]. Sanchez-Sanchez et al. [23] did not find associations between RS performance markers and Tc or Dm at DmMax on RF. In contrast, our results show, in accordance with Morales-Artacho et al. [17], that MCP changed in relation to the NMESa used. More concretely, we found a moderate association between RS performance markers and RF-Dm, but at submaximal NMESa (Figure 3). To our knowledge, only one previous study has assessed this theme [16]. The authors found, in agreement with our results, that a submaximal amplitude (i.e., 40 mA) showed the highest sensitivity to distinguish among jump or sprint performances in athletes (using the contraction velocity variable), also in the RF.

The muscle implication in different actions can explain the difference between the RF and BF associations during sprinting: while the BF has an important role in sprint performance in the horizontal force production, the RF has shown high electromyographic activation [26]. In addition, the difference in the response of both muscles can be explained by the Tc and Dm values found at lower and higher amplitudes. While the lowest Tc was found at 20 mA and the highest Tc at DmMax in BF, RF showed the opposite response. However, we observed the lowest Dm at 20 mA and the highest Dm at 100 mA in both muscles. Thus, we demonstrate a NMESa dependency on the MCP assessment, suggesting that those with a lower proportion of fast-twitch fibers and lower muscle stiffness (assuming the rationale justified in the introduction section) have worse performance when they perform RS.

Another important finding from our results (Figure 3 and Figure 4) shows the response of the submaximal and supramaximal MCP responses in the presence of fatigue (i.e., after a RS protocol) over a range of different NMESa. Again, we observed differences for each muscle. For the BF, the higher the NMESa amplitude, the higher association at +48 h between RS performance and ΔTc (i.e., 100 mA). This suggests that those with better RS performance experience a higher increase in Tc in BF. Previously, Lohr et al. [1] found inconsistencies in the Tc response after fatiguing tasks, suggesting that these responses can be attributed to general vs. local induced fatigue. More concretely, Sánchez-Sánchez et al. [23] did not find changes in the BF-Tc immediately after RS. It is important to note that in these studies, only the immediate response was analyzed (i.e., Post). Furthermore, we analyzed the time course of the responses up to +48 h, showing the highest associations at +48 h compared to post measurements, thus suggesting that the immediate transient fatigue of the BF after RS is not as high as 48 h after completing the exercise. The BF has an important role in horizontal force production [26]. The eccentric phase of the knee flexion during sprinting leads to increased hamstring muscle damage [27]. Our results showed an increment in CK at +24 h and +48 h after RS, suggesting an increase in muscle damage, which may explain the higher associations at +48 h between BF-Tc and RS performance. Furthermore, these responses can be explained by a decreased leg flexion peak torque at +48 h (Figure 2). In addition, we found that the RF had the highest negative associations between Dm and RS performance also at +48 h after the RS protocol, but at a submaximal amplitude (i.e., 40 mA). A decline in Dm can be explained by the impaired propagation of the electrical stimulus along the sarcolemma [22]. It can be suggested that this is shown in RF and not in BF after RS due to the muscle-specific recruitment during running [26].

The RS protocol used caused an increment in muscle damage (i.e., increase in CK levels) at +24 and +48 h. The increased CK concentration days after the protocol can be explained by the failure to fully activate the contracting muscle during multiple sprints [21], leading to a disruption of the cellular membrane, thus increasing the CK [28]. Previously, Hunter et al. [29] showed no relationship between CK concentration and MCP measured with TMG. The authors suggested that this could be related to the inter-individual variation in CK. Although our results also showed this variation (see the SD in Figure 2), we found high relationships between CK and MCP, especially in the BF, from post to +48 h, in agreement with Harmsen et al. [30]. However, we found these relationships with the lowest NMESa used (i.e., 20 mA). Thus, our results suggest that the lower the Dm, the higher the CK concentration after an RS protocol up to +48 h. Again, this can be explained by the implication of the fast-twitch fibers of the BF during sprinting [21], since the lower Dm could be attributed to the incapacity to recruit more fast-twitch fibers at lower NMESa. In addition, the leg flexion peak torque was reduced at +48 h. We found that, in both muscles, a Tc increment was associated with an MVIC increment, from Post to +48 h. In addition, in the BF, according to De Paula Simola et al. [31], a decrement in Dm was moderately associated with a decrement in MVIC. Interestingly, again, a submaximal NMESa showed the highest associations (i.e., 20 mA).

Caution should be taken when considering our results due to some limitations. First, participants were not experts in sprinting actions. Secondly, we used both legs from each participant for the analyses. However, no differences were previously found in MCP properties when the dominant and the non-dominant legs were studied using a similar RS protocol [23]. With our results, we provide further novel information about the potential usage of submaximal amplitudes to analyze the MCP using TMG. However, many of the mechanisms which may explain our results are still unknown. Further research should focus on studying the relationships between MCP measured and different NMESa and (1) the electromyographic muscle response, especially concerning the muscle conduction velocity, or (2) the histochemical distribution of motor unit types. In addition, it might be interesting to analyze if these findings also occur with different sports tasks (i.e., jumps, throws, or kicks) and with a larger and homogeneous sample size.

## 5. Conclusions

In conclusion, the stimulation amplitude determines the MCP during the tensiomyographic assessment. What is more, the MCPs are not only dependent on the NMESa, but also on the level of fatigue of the muscle studied. Lower submaximal amplitudes (i.e., 20–40 mA) may better reflect the fast-twitch fiber activity, but this is also dependent on the distribution of muscle fibers in each muscle. Thus, the time-consuming TMG application can be reduced for assessing multiple muscles during daily routines. Finally, to assess the RS performance in relation to the neuromuscular performance, RS-Loss is a valid option to assess the neuromuscular fatigue in RS, due to the associations found with the RF, either before or after performance of the RS protocol.

## Figures and Tables

**Figure 1 ijerph-18-11689-f001:**
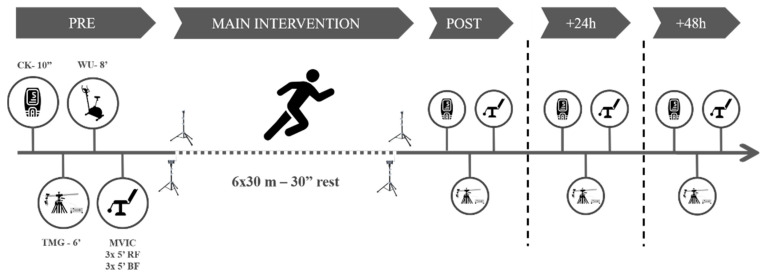
Study procedures. CK = creatine kinase. WU = warm up. TMG = tensiomyography. MVIC = maximum voluntary isometric contraction test.

**Figure 2 ijerph-18-11689-f002:**
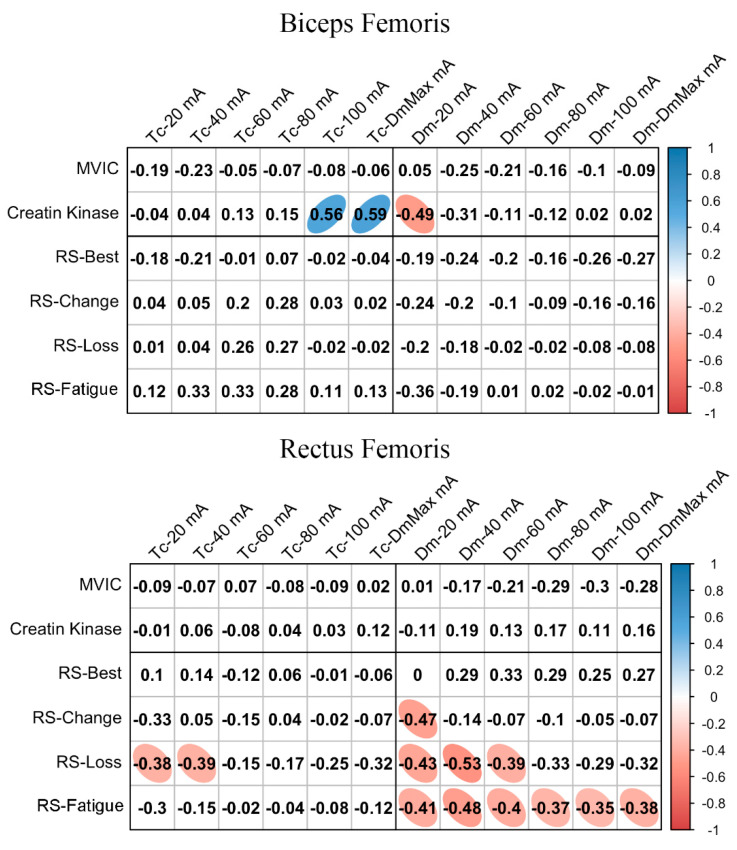
Basal contraction time (Tc) and muscle belly displacement (Dm) associations with repeated sprint (RS) fatigue and performance markers at all the electrical stimulation amplitudes used. Blue circles represent significant positive correlations, while red circles represent significant negative correlations (*p* < 0.05). The strength of the blue or red colors shows the strength of the correlation, also shown from 0 to 1 on the right bar. MVIC = peak torque during maximum voluntary isometric contraction. mA = milliamperes. DmMax = supramaximal amplitude stimulation.

**Figure 3 ijerph-18-11689-f003:**
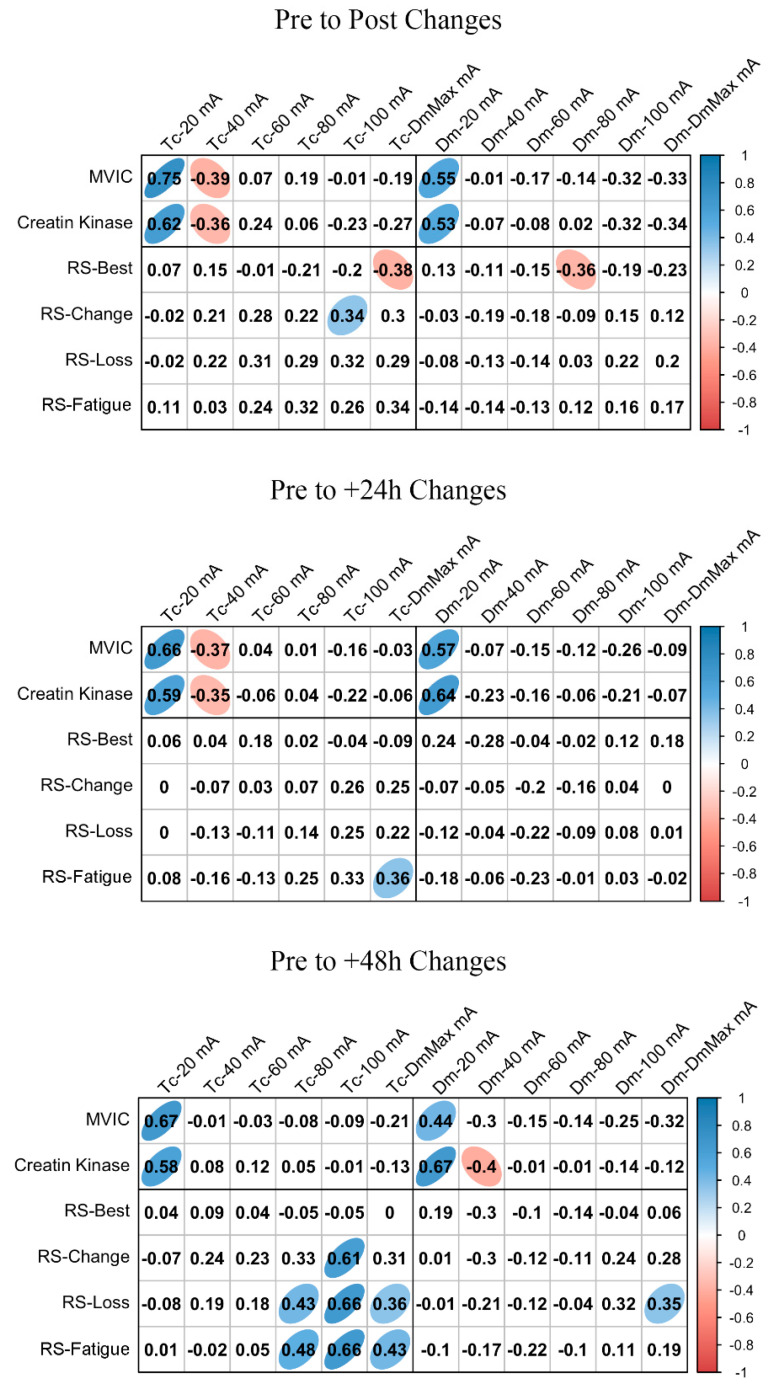
Biceps femoris associations between changes in contraction time (Tc) or muscle belly displacement (Dm), and repeated sprint (RS) fatigue changes and performance markers at all the electrical stimulation amplitudes used. Blue circles represent significant positive correlations, while red circles represent significant negative correlations (*p* < 0.05). The strength of the blue or red colors shows the strength of the correlation, also shown from 0 to 1 on the right bar. MVIC = peak torque during maximum voluntary isometric contraction. mA = milliamperes. DmMax = supramaximal amplitude stimulation.

**Figure 4 ijerph-18-11689-f004:**
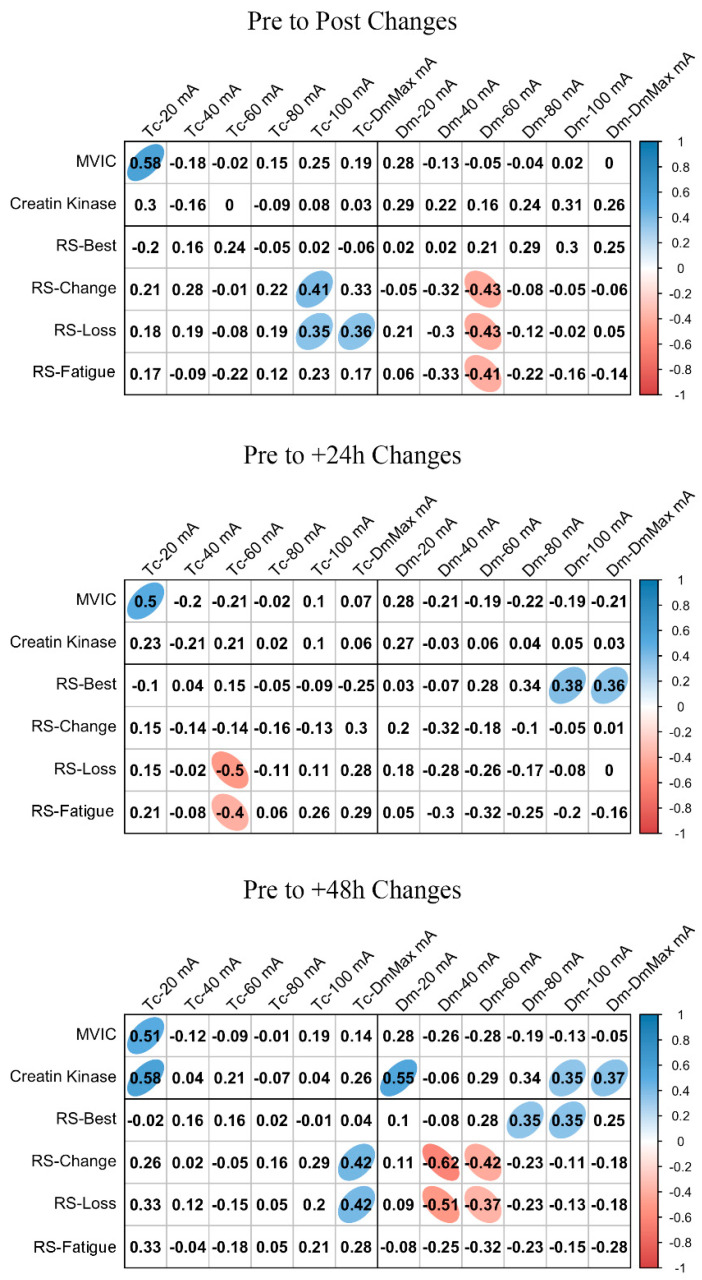
Rectus femoris associations between changes in contraction time (Tc) or muscle belly displacement (Dm), and repeated sprint (RS) fatigue changes and performance markers at all the electrical stimulation amplitudes used. Blue circles represent significant positive correlations, while red circles represent significant negative correlations (*p* < 0.05). The strength of the blue or red colors shows the strength of the correlation, also shown from 0 to 1 on the right bar. MVIC = peak torque during maximum voluntary isometric contraction. mA = milliamperes. DmMax = supramaximal amplitude stimulation.

**Table 1 ijerph-18-11689-t001:** Contraction time and muscle displacement time course descriptive values at different time course points.

Intensity	20 mA	40 mA	60 mA	80 mA	100 mA	Dm-Max mA	Amplitude Effect	
							*p*-Value	η^2^_p_
*Contraction Time (ms)*								
*Biceps Femoris*								
Pre	29.1 ± 13.6 ef	31.6 ± 10.3	33.1 ± 11.7	36.5 ± 11.9	40.1 ± 13.6 a	40.5 ± 13.5 a	<0.001	0.16
Post	37.08 ± 12.8 f	31.3 ± 14.8 f	35.4 ± 12.2 f	37.0 ± 12.6 f	41.7± 14.2	44.1 ± 13.5 abcd	<0.001	0.14
+24 h	38.3 ± 15.5	30.2 ± 15.6	34.3 ± 12.0 f	35.7 ± 15.0	40.2 ± 16.2	43.3 ± 16.7 c	0.003	0.12
+48 h	35.9 ± 13.6	31.5 ± 16.2	33.0 ± 11.9	34.2 ± 13.2	37.1 ± 16.2	39.6 ± 16.3	0.119	0.05
*Rectus Femoris*								
Pre	34.1 ± 9.6 *	29.5 ± 6.2 adef	27.7 ± 5.8 a	27.0 ± 4.3 abf	26.3 ± 4.3 abd	27.3 ± 5.1 ab	<0.001	0.35
Post	36.1 ± 13.5 ef	34.6 ± 8.5 def	31.9 ± 5.7 de	28.2 ± 6.4 bdf	27.6 ± 3.9 abc	26.9 ± 3.8 abc	<0.001	0.23
+24 h	35.6 ± 14.3	34.5 ± 9.5 def	30.8 ± 6.3 ef	29.1 ± 5.1 bef	28.1 ± 4.5 bcdf	27.4 ± 4.4 bcde	0.002	0.18
+48 h	32.1 ± 11.4	34.3 ± 9.8 def	31.1 ± 7.0 def	28.5 ± 6.0 bc	27.6 ± 4.6 bc	27.0 ± 4.2 bc	0.006	0.14
*Muscle Displacement (mm)*								
*Biceps Femoris*								
Pre	1.2 ± 1.3 *	3.7 ± 1.7 *	5.2 ± 2.1 *	6.7 ± 2.7 *	8.1 ± 2.7 abcd	8.1 ± 2.7 abcd	<0.001	0.82
Post	7.5 ± 3.8 bc	1.4 ± 1.4 *	3.9 ± 1.5 *	5.3 ± 2.0 bcef	6.9 ± 2.7 bcdf	8.1 ± 2.8 bcde	<0.001	0.60
+24 h	7.1 ± 3.9 bcd	1.3 ± 1.3 *	3.2 ± 1.5 *	4.4 ± 1.9 *	5.7 ± 2.5 bcdf	6.7 ± 2.9 bcde	<0.001	0.52
+48 h	7.6 ± 4.2 bc	1.6 ± 1.8 *	4.1 ± 2.4 *	5.3 ± 2.8 bcef	6.4 ± 2.3 bcd	7.0 ± 3.4 bcd	0.001	0.49
Rectus Femoris								
Pre	3.3 ± 2.1 *	7.4 ± 3.0 *	8.6 ± 2.8 *	9.2 ± 2.9 *	9.7 ± 3.1 abcd	10.1 ± 3.0 abcd	<0.001	0.80
Post	7.9 ± 3.6 b	4.3 ± 2.7 *	7.9 ± 2.8 bdef	8.7 ± 2.7 bcef	9.1 ± 2.6 bcd	9.5 ± 2.7 bcd	<0.001	0.44
+24 h	7.5 ± 3.9 b	4.0 ± 2.8 *	7.7 ± 3.0 bdef	8.6 ± 2.9 bcef	9.1 ± 2.9 bcdf	9.5 ± 3.0 bcde	<0.001	0.43
+48 h	7.0 ± 3.9 b	3.9 ± 2.7 *	2.2 ± 3.0 *	8.0 ± 2.9 bcef	8.6 ± 2.9 bcd	9.2 ± 2.9 bcd	<0.001	0.42

* = significant (*p* < 0.05) differences with all NMES amplitudes. Letters show a significant (*p* < 0.05) difference with a specific NMES amplitude (a = 20 mA; b = 40 mA; c = 60 mA; d = 80 mA; e = 100 mA; f = Dm-Max (maximum muscle displacement NMES amplitude). η^2^_p_ = partial eta-squared effect size.

## Data Availability

Data available upon request.

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
