# Peer review of "Muscle Contractile Properties Measured at Submaximal Electrical Amplitudes and Not at Supramaximal Amplitudes Are Associated with Repeated Sprint Performance and Fatigue Markers"

_ijerph, 2021, doi:10.3390/ijerph182111689_

Round 1
Reviewer 1 Report
In the article “Muscle Contractile Properties Measured at Submaximal Electrical Amplitudes and Not at Supramaximal Amplitudes are Associated with Repeated Sprint Performance and Fatigue Markers” Muñoz-López and colleagues demonstrated that tensiomyography performed at submaximal neuromuscular electrical stimulation amplitudes (NMESa) assessed better the muscle contractile properties (i.e: neuromuscular fatigue) after repeated sprints, with important practical applications. However, before the publication in International Journal of Environmental Research and Public Health, the authors should address some modifications.
Major points:
- Results 3.2 ‘MVIC and fatigue time course’ in Figure 2 there are the basal Tc and Dm associations in both muscles, I don’t see these data in that figure, maybe you might explain better.
- Line 207 ‘large negative association between Tc and 207 CK at high NMESa (100mA or DmMax).’ From the figure 2 (blu circle) the correlation is positive and not negative.
- Line 221 the muscle where you observe ‘positive associations between ΔDm and ΔMVIC at Post (large), +24h (large), and +48h (moderate)’ from the figure 2 is the BF not RF.
- Line 242 ‘Figures 4 and 5’ I don’t have figure 5 in the present version of the manuscript, so maybe you want to write 3 and 4.
- The same in line 387 ‘Figures 4 and 5’.
Minor points:
- Line 24 is the first time you use TMG please write in full.
- Line 68 please check if the word “properties” is necessary after MCP.
- Line 122 is the first time you use RF please write in full.
- Line 246 “were found between ΔTc and RS-change (moderate) or RS-Loss 245 (moderate), but at Post and at +48h” you might add that these associations were observed at high amplitudes.
Reviewer 2 Report
Review of the manuscript -Manuscript ID: ijerph-1427674
An interesting report was presented in the paper.
The survey is generally well written.
The title encourages you to read the content of the article.
The summary is complete and meets the requirements.
The aim of the publication and the hypothesis were clearly defined.
The Method section presents some shortcomings.
The group of respondents is not large, which the authors rightly emphasize in the limitations and conclusions of the study.
The characteristics of the volunteers in terms of age, height, weight, namely: "(age 23.2 [9.8] years, height 1.75 [0.01] m, body mass 72.7 [21.0] kg)", is not obvious. Is it mean, e.g. age and standard deviation (mean age 23.2 ± 9.8), etc.?
If this is a standard deviation, it seems that the selection of volunteers in such a small sample should be more precise, especially the body weight as well as the age of the respondents.
In my opinion, the sample size is missing, and the authors do not precisely describe the process of recruiting volunteers, as well as the inclusion and exclusion criteria for the study.
Likewise, the description of the procedure is not clear enough for the test to be repeated. Besides, it is not obvious why the authors wrote "following a protocol published elsewhere" and did not explain where?
The results of the study confirm the assumptions of the work.
Discussion_ is sufficiently supportive of the research results. The conclusions would be clearer if they were not included in the form of a description.
Taken together, these studies are important because, as the authors themselves write, their findings provide further evidence on the potential use of submaximal neuromuscular electrical stimulation amplitudes to analyze muscle contraction using tensiomigraphy. However, the research requires clarification and confirmation on a large, homogeneous group of participants.
The work requires corrections before it is allowed to be published.
Reviewer 3 Report
The authors have done a fine job on the manuscript. I believe it clear and well cited. I only have a few minor things to add/suggest:
Line 23: Instead of saying “important associations”, I recommend authors use the term “lack of practical significance” or something to that degree.
Line 35: example of high? Only an example of low was given.
Line 71: increases. Check for typos once more throughout
The intro is written nicely
Line 99: remove “sports science students”
Line 102: Were there any exclusions for recent injury (i.e. the past 6 months)?
Line 112: Where did the 6 x 30 m protocol come from? Simply cite or explain.
Line 120: Are samples run in duplicate? I am not familiar with this analyzer so more details would be appreciated by the reader, I think.
Line 152: The pictures are very small and hard to see. Can they be enlarged some?
Line 232: I very much like the way authors have shown the correlations with the scale and clear lines. However, the table prior to it is hard to follow. It may be simply because it runs on two pages. I would suggest ensuring it is on one page, or splitting it up somehow.
Authors did a great job on the discussion, but I think it need more practical implications included. What should practitioners do with this info?
Line 428: If time of day was not controlled for, it should be listed as a limitation here.
Round 2
Reviewer 1 Report
In the revised version of the manuscript Muñoz-López and colleagues modified appropriately the text as requested, improving the quality of the paper.